# Evaluation of Deep Reinforcement Learning Methods for Modular Robots

**Risto Kojcev, Nora Etxezarreta, Alejandro Hernández and Víctor Mayoral**
Erle Robotics
Venta de la Estrella Kalea, 6
01006 Vitoria-Gasteiz, Araba, Spain
`{risto, nora, alex, victor}@erlerobotics.com`

## Abstract

We propose a novel framework for Deep Reinforcement Learning (DRL) in modular robotics using traditional robotic tools that extend state-of-the-art DRL implementations and provide an end-to-end approach which trains a robot directly from joint states. Moreover, we present a novel technique to transfer these DLR methods into the real robot, aiming to close the simulation-reality gap. We demonstrate the robustness of the performance of state-of-the-art DRL methods for continuous action spaces in modular robots, with an empirical study both in simulation and in the real robot where we also evaluate how accelerating the simulation time affects the robot's performance. Our results show that extending the modular robot from 3 degrees-of-freedom (DoF), to 4 DoF, does not affect the robot's learning. This paves the way towards training modular robots using DRL techniques.

## 1    Introduction

Current robot systems are designed, built and programmed by teams with multidisciplinary skills. The traditional approach to program such systems is typically referred to as the robotics control pipeline and requires going from observations to final low-level control commands through: a) state estimation, b) modeling and prediction, c) planning, and d) low level control translation. As introduced by Zamalloa et al. (2017), the whole process requires fine tuning of every step in the pipeline incurring into a relevant complexity where optimization at every step is critical and has a direct impact in the final result.

In recent years, several techniques for DRL have shown good success in learning complex behaviour skills and solving challenging control tasks in high-dimensional state-space (Levine & Koltun, 2013; Peters & Schaal, 2008; Schulman et al., 2015; 2017; Wu et al., 2017). However, many of the benchmarked environments, such as Atari (Mnih et al., 2013) and Mujoco (Todorov et al., 2012), rarely deal with realistic or complex environments (Nogueira et al., 2017; Zamora et al., 2016), or utilize the tools commonly used in robotics such as the Robot Operating System (ROS) (Quigley et al., 2009). The research conducted in the previous work can only be translated into real world robots with a considerable amount of effort for each particular robot. Thus, the scalability of previous methods for modular robots is questionable.

Modular robots can extend their components seamlessly. This brings clear advantages for their construction, however, training them with current DRL methods becomes cumbersome due to the following reasons: every small change in the physical structure of the robot will require a new training, building the tools to train modular robots (such as the simulation model, virtual drivers) is a time consuming process, and transferring the results to the real robot is complex given the flexibility of these systems.

In this work we present a framework that employs the traditional tools in the robotics field, such as Gazebo (Koenig & Howard (2004)) and ROS, which simplifies the process of building modular robots and their corresponding tools. Our framework includes baseline implementations (Dhariwal et al., 2017) for the most common DRL techniques dealing with policy iteration methods. Using this framework, we present configurations with 3 and 4 degrees-of-freedom (DoF), while performing the

same task. In addition, we introduce our insights about the impact of the simulation acceleration in the final reward.

## 2   PREVIOUS WORK

DRL methods have shown great success when dealing with high-dimensional, continuous state and action spaces found in robotics. For our experiments, we focus on the DRL methods that have shown best performance and highest robustness against different environments and hyperparameter configurations, namely the Proximal Policy Optimization (PPO) Schulman et al. (2017) methods. In a nutshell, PPO alternates between sampling data trough interaction with the environment and optimizing the 'surrogate' objective by clipping the policy probability ratio.

Previous work focused on the simulation-to-reality transfer problem, Barrett et al. (2010); Rusu et al. (2016); James & Johns (2016), presents partial success of transferring learned behavior in simulation to a real robot. These works explain the importance of having scenes in simulation as similar as possible to the reality in order to simplify the process of transferring the learned behavior to real scenarios. Zhu et al. (2017) describe a high-quality and realistic 3D scenes. The approach of Tobin et al. (2017) randomizes the rendering in simulation, reaching enough variability in the simulator. This allows for the images in the real world to be considered as just another variation in the simulator. To the best of our knowledge, the work conducted in previous approaches focuses on restricted scenarios in a controlled environment, where specific algorithms for solving particular task were used. This is not the case when a robotic system needs to be deployed in realistic scenarios, specially if the robot is modular and can therefore present a number of different configurations.

## 3   PRELIMINARY RESULTS

### 3.1   EXPERIMENTAL SETUP

As previously presented in Zamora et al. (2016), our novel technique for transferring any network trained in simulation using DRL techniques to the real robot relies on our extension of the OpenAI gym which is tailored for robotics. For our experiments, we train two modular robots, namely the SCARA 3DoF and 4DoF robots, where the Gazebo simulator and corresponding ROS packages convert the actions generated from each algorithm to appropriate trajectories the robot can execute.

The initial position of the robot is set to zero for all joints. The reward is modeled as Residual Mean Square Error (RMSE) between the current position of the end-effector and the goal. The goal is set to be in a selected point in the environment, particularly, the center of the **H** letter in the workspace of the robot. This translates to coordinates $[0.3305805, -0.1326121, 0.3746]$ for the 3DoF Scara robot and $[0.3305805, -0.1326121, 0.4868]$ for the 4DoF Scara robot, with respect to the origin of the environment, which in our case is set to be the base of the robot. The range of the reward is set to be between $[-1, 1]$. The robot gets a positive reward when the RMSE is smaller than $0.005$ and negative reward otherwise. The robot is reset to the initial position when RMSE is smaller than $0.005$, or when the number of steps exceeds the maximum timesteps for an episode.

### 3.2   EXPERIMENTAL RESULTS

We have evaluated how the trajectory execution time influences the reward of PPO1 and PPO2 during training as shown in Figure 3.2. We have evaluated PPO1 and PPO2 methods with a trajectory execution time of $1s$, $100ms$, $10ms$ or $1ms$. Figure 4 illustrates the recorded trajectories when executing previously trained behaviour to the real 3DoF and 4DoF modular Scara robot and Table 1 summarizes the results of the Euclidean Distance, given in millimeters, between reached end-effector position and the real target. As we can observe from the obtained results, for PPO1 and PPO2 the 3DoF robot has best performance when the training time is set to $1ms$. On the other hand, when the trajectory execution time is set to $1s$, PPO1 and PPO2 have worst performance. In the case of the 4DoF robot, PPO1 shows best performance when the trajectory execution time

is set to $10ms$ and worst performance when the trajectory execution time is set to $100ms$. On the other hand, PPO2 for the 4DoF Scara has best performance when the simulation time is set to $1ms$, and worst performance when the trajectory execution time is $1s$. Accelerating simulation time allows PPO1 and PPO2 to converge faster as they need a lower number of time steps. As a result, faster convergence reduces the training time while preserving performance. We can conclude that training networks with accelerated trajectory execution times provides equal or even better results than training the robot in real-time.

| | Method | Euclidean Distance (mm) vs. simulation time | | | |
| --- | --- | --- | --- | --- | --- |
| | | $1s$ | $100ms$ | $10ms$ | $1ms$ |
| **3DoF** | PPO1 | 52.47±0.11 | 44.18±0.13 | 21.3±0.01 | **13.09±0.06** |
| | PPO2 | 317.44±0.08 | 69.08±0.13 | 189.09±0.21 | **23.63±0.21** |
| **4DoF** | PPO1 | 37.02±0.12 | 248.48±0.04 | **20.33±0.23** | 105.74±0.07 |
| | PPO2 | 656.22±0.03 | 98.87±0.07 | 73.07±0.09 | **58.19±0.03** |

Table 1: Summarized results when executing a network trained with different trajectory execution times. The target is set to the middle of the H for the 3DoF and 4DoF robots.

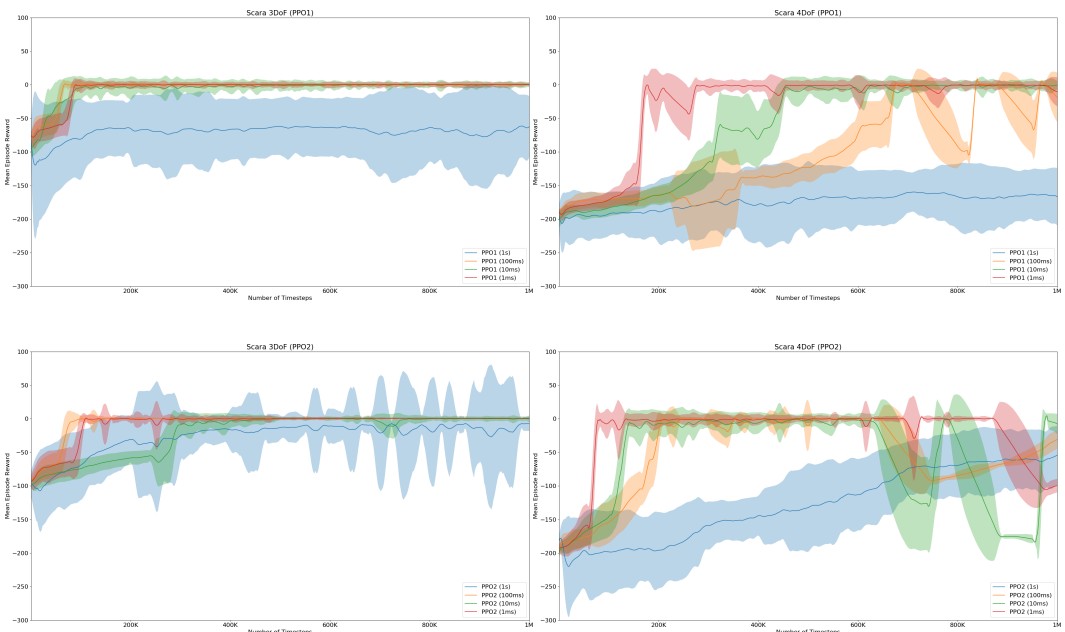

Figure 1: Mean Episode reward for training the 3DoF Scara robot with PPO1 (top left) and PPO2 (top right) and training the 4DoF Scara robot with PPO1 (bottom left) and PPO2 (bottom right) when executing trajectories with different times.

There still remain many challenges within the DRL field for robotics. The main problems are the long training times, the simulation-to-real robot transfer, reward shaping, sample efficiency and extending the behaviour to diverse tasks and robot configurations.

So far, our work with modular robots has focused on simple tasks like reaching a point in space. In order to have an end-to-end training framework (from pixels to motor torques) and to perform more complex tasks, we aim to integrate additional rich sensory input, such as vision. We envision the future of robotics to be about modular robots where the trained network can generalize online to modifications in the robot such as change of a component or dynamic obstacle avoidance.

## 4 APPENDIX

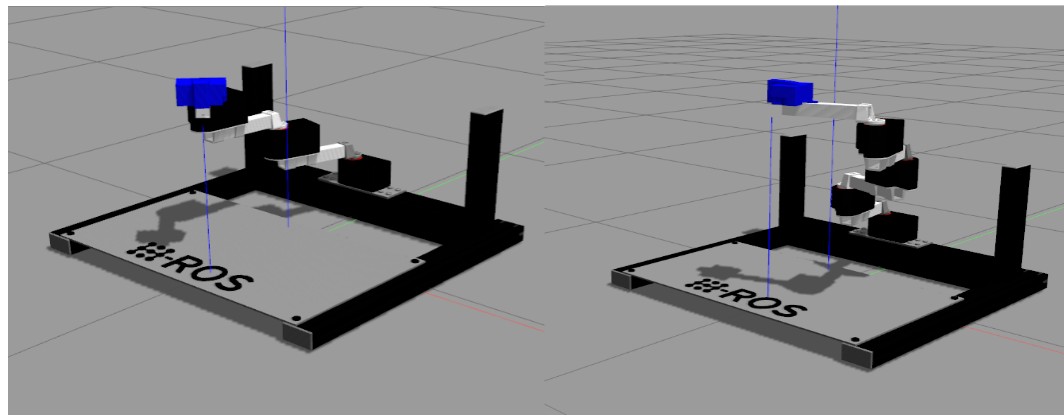

Figure 2: All the training for the 3DoF (illustrated on the left) and 4DoF (illustrated on the right) Scara robot is performed in simulation in our environment. Then, the trained network is transferred to the real robot.

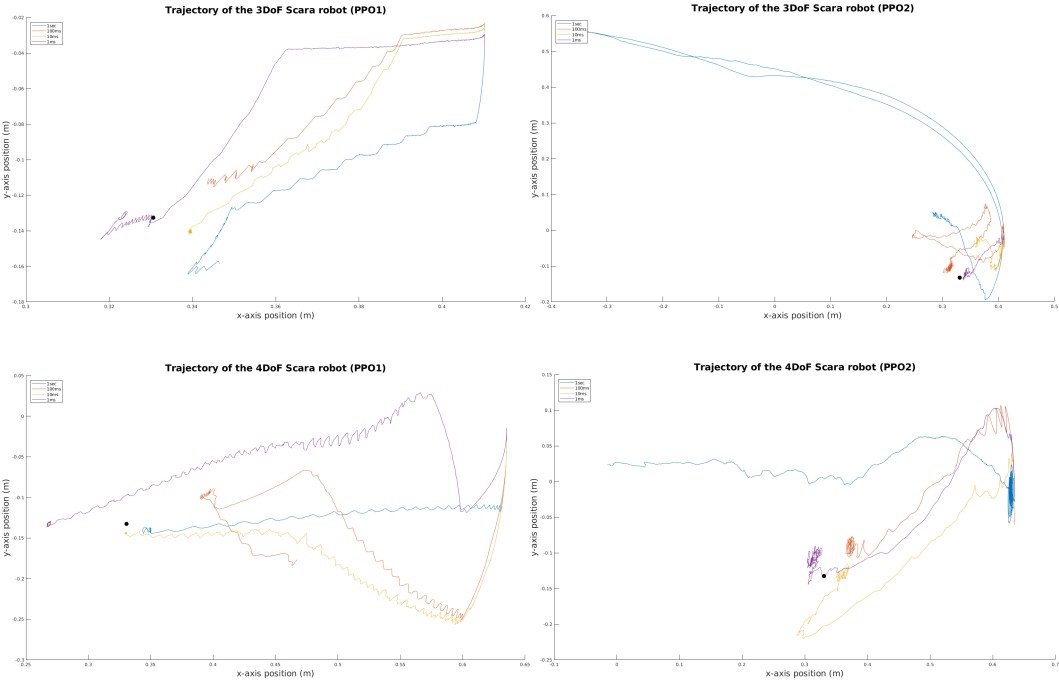

Figure 3: Output of the trajectories for the 3DoF (top) and 4DoF (bottom) Scara Robot, when loaded to a previously trained network for different amounts of simulation time.

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

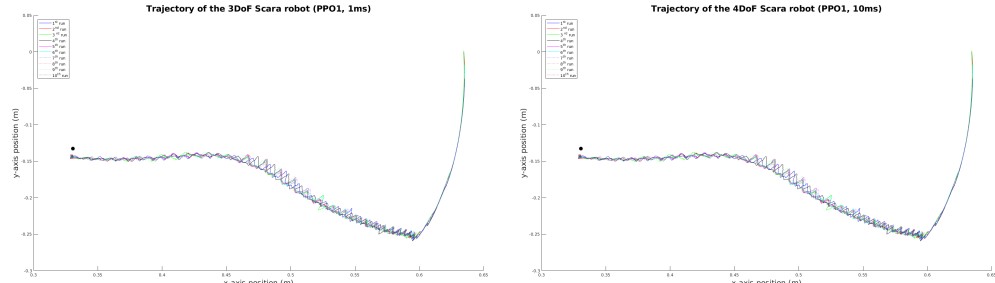

Figure 4: Output of the trajectories for the 3DoF Scara Robot (left) when reaching the target point with a network trained with $1ms$ trajectory execution time. On the right, the recorded trajectories of the 4DoF Scara robot, trained with the PPO1 method and trajectory execution time of $10ms$. The differences in the recorded trajectories is due to inaccuracies of the motor driver when going trough certain point of the trajectory.

Stephen James and Edward Johns. 3d simulation for robot arm control with deep q-learning. CoRR, abs/1609.03759, 2016. URL http://arxiv.org/abs/1609.03759.

Nathan Koenig and Andrew Howard. Design and use paradigms for gazebo, an open-source multi-robot simulator. In Intelligent Robots and Systems, 2004.(IROS 2004). Proceedings. 2004 IEEE/RSJ International Conference on, volume 3, pp. 2149–2154. IEEE, 2004.

Sergey Levine and Vladlen Koltun. Guided policy search. In Proceedings of the 30th International Conference on Machine Learning (ICML-13), pp. 1–9, 2013.

Volodymyr Mnih, Koray Kavukcuoglu, David Silver, Alex Graves, Ioannis Antonoglou, Daan Wierstra, and Martin Riedmiller. Playing atari with deep reinforcement learning. arXiv preprint arXiv:1312.5602, 2013.

T. Nogueira, S. Fratini, and K. Schilling. Autonomously controlling flexible timelines: From domain-independent planning to robust execution. In 2017 IEEE Aerospace Conference, pp. 1–15, March 2017. doi: 10.1109/AERO.2017.7943603.

Jan Peters and Stefan Schaal. Reinforcement learning of motor skills with policy gradients. Neural networks, 21(4):682–697, 2008.

Morgan Quigley, Ken Conley, Brian Gerkey, Josh Faust, Tully Foote, Jeremy Leibs, Rob Wheeler, and Andrew Y Ng. Ros: an open-source robot operating system. In ICRA workshop on open source software, volume 3, pp. 5. Kobe, 2009.

Andrei A. Rusu, Matej Vecerik, Thomas Rothörl, Nicolas Heess, Razvan Pascanu, and Raia Hadsell. Sim-to-real robot learning from pixels with progressive nets. CoRR, abs/1610.04286, 2016. URL http://arxiv.org/abs/1610.04286.

John Schulman, Sergey Levine, Pieter Abbeel, Michael Jordan, and Philipp Moritz. Trust region policy optimization. In Proceedings of the 32nd International Conference on Machine Learning (ICML-15), pp. 1889–1897, 2015.

John Schulman, Filip Wolski, Prafulla Dhariwal, Alec Radford, and Oleg Klimov. Proximal policy optimization algorithms. arXiv preprint arXiv:1707.06347, 2017.

Josh Tobin, Rachel Fong, Alex Ray, Jonas Schneider, Wojciech Zaremba, and Pieter Abbeel. Domain randomization for transferring deep neural networks from simulation to the real world. arXiv preprint arXiv:1703.06907, 2017.

Emanuel Todorov, Tom Erez, and Yuval Tassa. Mujoco: A physics engine for model-based control. In Intelligent Robots and Systems (IROS), 2012 IEEE/RSJ International Conference on, pp. 5026–5033. IEEE, 2012.

Yuhuai Wu, Elman Mansimov, Roger B Grosse, Shun Liao, and Jimmy Ba. Scalable trust-region method for deep reinforcement learning using kronecker-factored approximation. In Advances in Neural Information Processing Systems, pp. 5285–5294, 2017.

Irati Zamalloa, Risto Kojcev, Alejandro Hernández, Iñigo Muguruza, Lander Usategui, Asier Bilbao, and Víctor Mayoral. Dissecting robotics-historical overview and future perspectives. arXiv preprint arXiv:1704.08617, 2017.

Iker Zamora, Nestor Gonzalez Lopez, Victor Mayoral Vilches, and Alejandro Hernandez Cordero. Extending the openai gym for robotics: a toolkit for reinforcement learning using ros and gazebo. arXiv preprint arXiv:1608.05742, 2016.

Yuke Zhu, Roozbeh Mottaghi, Eric Kolve, Joseph J Lim, Abhinav Gupta, Li Fei-Fei, and Ali Farhadi. Target-driven visual navigation in indoor scenes using deep reinforcement learning. In Robotics and Automation (ICRA), 2017 IEEE International Conference on, pp. 3357–3364. IEEE, 2017.

