# OpenReview forum: "Evaluation of Deep Reinforcement Learning Methods for Modular Robots"
_ICLR.cc/2018/Workshop — Reject_

### Official Review · AnonReviewer1 · 2018-03-06
**Incomplete submission?**

**Rating:** 3
**Confidence:** 4

**Review:**

This paper is either incomplete or poorly written. It claims a lot in the abstract and introduction, such as novel technique in sim-to-real transfer or accelerating simulation times. However, I did not find any methods or discussions related to these claims. The method section is completely missing.

Although the "modular" robot seems to be a big focus of this paper, it is not clear to me what part of the algorithm (which is missing) or the results are specific to improve learning for "modular" robots.

The results section is not clear too. For example, the terms PPO1 and PPO2 are not defined. I am not sure what their differences are. The task in the evaluation is a very simple reaching task. And I am not sure what is the conclusion of these preliminary results.

In summary, I think that this submission is not complete. I would not recommend accepting it for now.

---

### Official Review · AnonReviewer3 · 2018-03-09
**Application of RL to modular robotics, nothing really novel or particularly interesting here**

**Rating:** 1
**Confidence:** 5

**Review:**

This paper presents an applications of RL techniques to train a modular robot system with 3 or 4 degrees of freedom, aimed at a simple movement task.  The paper ultimately just seems to show that an RL policy (again, just for moving the robot) can be trained in simulation and then executed on the real system.

This paper is clearly not appropriate for the workshop track, and I think the authors may have misunderstood the purpose of this track.  As highlighted in the call, the workshops track is for "late-breaking developments, very novel ideas and position papers."  This paper is none of these, it is a simple demonstration of existing approaches to a modular robot system with no real experimental design or scientific interest.

I realize this review may come across as overly harsh, but this paper is just clearly unsuited for the ICLR workshops, so little additional consideration is needed, I believe.

---

### Official Review · AnonReviewer2 · 2018-03-10
**Promise of comparison between DRL methods, but nowhere to be found**

**Rating:** 3
**Confidence:** 4

**Review:**

- The task in this paper appears extremely simple (reach a fixed point). This begs the question if the generalization to modular robots with more modules (3 DOF to 4 DOF) is valid? It seems that the task is just too simple and can be solved with both setups. It would be more interesting to see a task for which the performance increases with increasing modularity as it would demonstrate that the algorithm can exploit the added capabilities of more complex robots.
- No comparison between methods. The title mentions DRL methodS, but I only see one algorithm (PPO)
- PPO1 vs PPO2: What's the difference
- The paper doesn't really have a conclusion, it just ends after the experiment.

---

### Decision · Program_Chairs · 2018-03-20
**ICLR 2018 Workshop Acceptance Decision**

**Decision:**

Reject

**Comment:**

Based on the reviews, this paper has not been accepted for presentation at the ICLR workshop. However, the conversation and updates can continue to appear here on OpenReview.